# Engaging Health and Aged Care Workers in Rural and Remote Australia Around Factors Impacting Their Access to and Participation in Dementia Training

**DOI:** 10.3390/geriatrics10010028

**Published:** 2025-02-14

**Authors:** Sandra C. Thompson, Jessica Valentine, Kira Gusterson, Katrina P. Fyfe, Alex Beilby, John A. Woods, Myles Clarkson Fletcher, Pascale Dettwiller, Kathryn W. Fitzgerald

**Affiliations:** 1Western Australian Centre for Rural Health (WACRH), University of Western Australia, 167 Fitzgerald St., Geraldton, WA 6530, Australia; kathryn.fitzgerald@uwa.edu.au; 2Western Australian Centre for Rural Health (WACRH), School of Allied Health, University of Western Australia, 35 Stirling Highway, Crawley, Perth, WA 6009, Australia; jessica.valentine@uwa.edu.au (J.V.); katrina.fyfe@uwa.edu.au (K.P.F.); john.woods@uwa.edu.au (J.A.W.); 3Western Australian Centre for Rural Health (WACRH), 66 Welcome Road, Karratha, WA 6714, Australia; kira.gusterson@hopecs.org.au; 4Dementia Training Australia WA, University of Western Australia, Perth, WA 6009, Australia; 5Western Australian Centre for Rural Health (WACRH), University of Western Australia, 22 Cleaver St., Carnarvon, WA 6701, Australia; alex.beilby@health.wa.gov.au; 6Western Australian Country Health Services, Albany Hospital, Warden Avenue, Spencer Park, Albany, WA 6330, Australia; 7Centre for Rural Health, University of Tasmania, 12 The Avenue, New Norfolk, TAS 7140, Australia; myles.clarksonfletcher@utas.edu.au; 8Department of Rural Health, University of South Australia, 86 Tasman Terrace, Port Lincoln, SA 5606, Australia; pascale.dettwiller@unisa.edu.au

**Keywords:** dementia, cognitive impairment, rural, quality of care, management, training, education, access, aged care, professional development, barriers, enablers

## Abstract

**Objective:** To better understand barriers and enablers to uptake of dementia training in rural and remote areas using input from rural and remote aged and health care workers into how dementia training could be offered to better meet their needs. **Methods**: Roundtable focus groups were conducted in six diverse rural and remote locations in four jurisdictions around Australia. Sixty-seven workers from predominantly nursing, allied health, and support worker roles involved in dementia care participated. Data were collected by site and used a mixture of face-to-face and virtual facilitated ‘roundtable’ discussions. Each group discussed barriers and enablers to participation in training and their preferences for how dementia training should be provided. **Results**: Commonalities emerged in barriers for accessing dementia training. Participants emphasised the need for strong organisational support and locally relevant, interactive and flexible delivery methods to address rural challenges. Significant challenges related to staffing levels, time constraints, and competing priorities. Enablers of training uptake included support from employers covering time and costs of training, local collaboration, and training accessibility for all job roles rather than profession specific. Participants emphasised the importance of practical, local training relevant to their scope of practice delivered by experienced trainers. The need for culturally safe aged care practices was noted in all sites. **Discussion and Conclusions**: Collaborative approaches across organisations and the aged care workforce and training relevant to local rural contexts were favoured. The opportunity to learn from external experts was greatly appreciated. Workers want training that enhances culturally safe practices. Organisational support is critical for training implementation.

## 1. Introduction

Dementia is a significant and growing health and aged care issue in Australia and around the world. The condition is responsible for the greatest burden of disease of any chronic illness in older Australians [1], reflecting healthy years of life lost due to living with ill health (non-fatal burden) plus the years of life lost due to dying prematurely (fatal burden) [2]. Dementia rises rapidly in prevalence after 65 years and dementia is the leading cause of death for women over 65 years and for men and women over 85 years [2]. Approximately one-third of Australians, including two of every five of those with dementia, live in regional, rural or remote areas where there are higher rates of dementia risk factors and the population is ageing due to emigration of younger adults [3,4]. Regional, rural, and remote areas are defined by their population size, distance to urban centres, and access to services. Regional towns or cities are usually service centres that have more and larger health services. Rural areas sit outside a regional centre, within a few hours’ drive. Remote refers to areas and towns far removed from a major capital or regional centre. Complicating the situation for dementia care in many rural and remote areas is the limited access to health care, including diagnostic, treatment and support services [5].

Dementia substantially impacts the health and quality of life of people with the condition, as well as their family and friends, and caring for people living with dementia requires a skilled and well-trained workforce. However, both in Australia and internationally, significant gaps exist in the attitudes, knowledge, and skills of health professionals regarding the care of people living with dementia [6,7,8]. In Australia, dementia education has been found to be minimal in medical school education curricula, and general practitioners report inadequate training in dementia care [9]. The recent Royal Commission into Aged Care Quality and Safety found dementia care to be an area requiring urgent attention, concluding that “the quality of dementia care in the aged care system needs significant and immediate improvement” [5]. The Commission recommended that all workers in aged care services undertake regular dementia training, as well as noting the need for regulation for mandatory minimum education standards and ongoing training requirements [5].

Despite higher rates of dementia in regional, rural and remote parts of Australia, health and aged care workers (HACW) in these areas face specific challenges in accessing dementia care training [10] and can have different professional development needs from their city counterparts [11]. Such challenges affecting the healthcare workforce in these communities can lead to (or exacerbate) unmet health needs and poor care for people living with dementia [10]. This situation is compounded for First Nations Australians requiring culturally safe dementia care [5].

Research surrounding the educational needs, barriers and enablers (factors) to accessing dementia training by health care professionals in regional, rural and remote settings is lacking [12], so more attention to the specific needs of rural and remote health and aged care workers with respect to dementia care is needed. Dementia Training Australia (DTA) UWA partnered with the Western Australian Centre for Rural Health (WACRH), a University Department of Rural Health (UDRHs), to explore the barriers and enablers to uptake of dementia training in rural and remote areas of Australia, to identify targeted training solutions, and to develop a rural and remote dementia training framework. WACRH is part of a network of UDRHs which are linked through the Australian Rural Health Education Network (AHREN). WACRH was one of several UDRHs funded for Aged Care Extension projects to undertake training in aged care in rural and remote sites. The current paper reports on findings from facilitated discussions (referred to as ‘roundtables’) conducted at several sites across rural and remote Australia, in which invited aged and healthcare workers (HACW) provided input into how training in dementia care can better meet their needs. The roundtable discussions aimed to inform the development of targeted and practical rural and remote dementia training, primarily by enhancing understanding of the workforce uptake of training in this setting. The roundtable participants also provided information on how the mode and content of dementia training could better meet their needs and described training approaches which they considered successful for the management of other conditions and that could be adapted to training for dementia care.

## 2. Materials and Methods

A qualitative exploratory approach was used to capture the views of HACW who work with people living with dementia in regional, rural, and remote areas. The group interview (‘roundtable’) was chosen over individual interviews for pragmatic reasons and also because it promotes the ability of participants to clarify their views, voice their opinions, and share their experiences that might not surface during individual interviews [13]. The approach for the focus group roundtables was developed specifically for the project through discussion within the WACRH investigation team to cover the areas of interest to be explored.

Identification of suitable participants was facilitated through the Australian Rural Health Education Network (ARHEN) aged care network. ARHEN is the national peak body for nineteen University Departments of Rural Health, of which WACRH is one. A key contact for each roundtable site was identified, and these contacts invited HACW from their local network through various means, including via email. All participants were informed of the research question, “What training and education approaches are preferred by rural and remote aged care providers for learning about dementia?” and given information about the research, including a participant information form, and provided written consent. Roundtable interactions were achieved through a ‘hybrid’ approach of face-to-face and virtual interactions to enable as many participants as possible to attend and ensure that location was not a barrier.

### The Roundtable Process

Six roundtables were held in June/July 2023 in diverse rural and remote locations across five Australian states (New Norfolk, Tasmania (Modified Monash (MM2)), Geraldton, Western Australia (WA) (MM3), Lismore/Northern Rivers, New South Wales (MM3/MM5), Chinchilla, Queensland (MM5), Carnarvon (WA) (MM6), Port Lincoln, South Australia (MM6)). The MM classification is used by government and defines whether a location is metropolitan, rural, remote, or very remote based on categories MM1 to MM 7. MM 1 is a major city and MM 7 is very remote. Sites were selected based on being recently funded to deliver training in aged care to nursing and allied health students under an extension to the Rural Health Multidisciplinary Training (RHMT) Program [14].

All roundtable discussions were facilitated by the same experienced health professional located in Geraldton, Western Australia (WA). Four of the six roundtables were facilitated remotely using Microsoft Teams; one was facilitated fully face-to-face, and one a combination of face-to-face and Microsoft Teams. The research team, who were in various WA locations, used several means to support the delivery and documentation of the interactions. This included utilising a PowerPoint presentation to guide the discussion, an electronic virtual whiteboard so that relevant information was iteratively captured and could be refined through further discussion during the session, hard copy materials that participants utilised in real-time as an activity, and video recording of the sessions via Microsoft Teams^®^ for further analysis later. Written material from small group activities was collated by the local site coordinator and sent to the research team for inclusion in the analysis. The researchers also communicated with each other throughout the sessions via the Teams^®^ chat function.

Each roundtable was ninety minutes. The key issues were explored through discussions and small group activities focussing on factors impacting access to and participation in dementia training in rural and remote areas experienced by participants, discussion to confirm and extend the barriers and enablers identified in a scoping review of the literature [12], and exploring successful experiences in any training that participants attended that may be able to be applied to dementia training.

Participants first discussed their personal experiences about what made it easy or difficult to attend dementia training in their rural or remote work setting. Priority factors were identified and confirmed. An overview of findings from the scoping review which examined barriers and enablers of uptake of dementia training in rural and remote health and aged workers in Australia and elsewhere was then presented [12]. Participants discussed and confirmed the barriers and enablers identified in the review and were invited to identify any additional factors that had not been discussed. To identify features of training which work well in rural and remote areas, participants were invited to describe experiences in any other training they had attended that could be adapted to dementia care. There was also an opportunity to discuss training needs for culturally safe aged care practice. As different considerations surfaced and were discussed, priorities were captured through a consensus approach.

Analysis proceeded simultaneously with data collection. After each of the Roundtables, the core group met to distil and extract the key issues identified, with summaries of each roundtable sent back to all participants post-meeting as a form of member checking to allow for any additions or further refinement to the issues captured. The findings of all Roundtables were then consolidated and grouped as themes, circulated to the wider group of investigators, and then overlaid by a small group with the framework of micro, meso and macro levels to differentiate individual, organisational and policy levels at which any potential issue or intervention would sit.

The research received approval from the Human Research Ethics Office at the University of Western Australia (2023/ET000245, approved 19 June 2023).

## 3. Results

### 3.1. Setting and Participants

A total of sixty-six health and aged care workers (HACWs) participated in this study, with six to sixteen participants in each roundtable from the following sites: Geraldton (WA) (*n* = 11), Carnarvon (WA) (*n* = 16), New Norfolk (Tas) (*n* = 8), Port Lincoln (SA) (*n* = 8), Lismore/Northern Rivers (NSW) (*n* = 9) and Chinchilla (Qld) (*n* = 14).

Health and aged care roles included: nurses (*n* = 27), allied health professionals (*n* = 15), personal carers/support workers (*n* = 6), management or administration (*n* = 15), and general practitioner (*n* = 1) (Table 1). Two participants did not specify their professional role. Seventy-six percent (76%) of participants worked in roles providing direct clinical or personal care to people living with dementia, and twenty-three percent (23%) were in management roles.

Table 2 shows that participants were mostly experienced in working with people living with dementia, with 42% of participants reporting more than ten years of experience. Forty-six percent (46%) of participants had worked in rural or remote areas for over ten years.

Themes emerged from the discussions and activities in relation to experiences with dementia training, as well as reflection and recommendations based on other training that participants had attended.

These themes were categorised into three levels (Table 3): individual (micro), workplace and training organisation (meso) and government/funding and regulatory (macro), reflecting the level at which interventions to improve the understanding of the aged care workforce were needed.

### 3.2. Individual Considerations (Micro)

The responses from participants that represented a micro perspective related to personal factors impacting on training uptake were grouped into six main themes: learning preferences and priorities; engagement and participation; personal interest and relevance; logistics and convenience; costs; and computers and technology.

#### 3.2.1. Learning Preferences and Priorities

While participants preferred face-to-face and ‘hands-on’ training, the flexibility of online courses was widely acknowledged, particularly in the context of geographical remoteness. Many noted the value of a combination of practical face-to-face training together with some online training, preferring interactive online opportunities as opposed to pre-recordings. Several mentioned the fatigue of online training following COVID-19.


*Many people are burnt out from online options following COVID-19 and therefore would be happy to travel (for example, over 4 h) with their peers to attend an in-person training in a central location.*
(Lismore/Northern Rivers)

It was recognised that training does not always require attendance in an organised course and that there may be opportunities within teams to share ideas and learn with and from colleagues. Participants highly valued being able to attend training with others, regardless of their roles both in their organisation and across organisations in the same area that provide care to those living with dementia.


*Training should be open to everyone, and inclusive of all staff such as cleaners, cooks, everyone who interacts with patients, and not just limited to clinicians.*
(Lismore/Northern Rivers)

The value of having input into the types of training attended was also repeatedly identified.

#### 3.2.2. Engagement and Participation

Several participants recalled training sessions they had attended that were ‘emotive’ or ‘immersive’, noting these as memorable and more valuable. For example, one participant recalled:


*…training with an immersive experience and being in the shoes of somebody with dementia. It provoked different emotions, such as complete fear and helplessness, which aided in understanding the feelings of patients living with dementia. It helps explain some of the responsive behaviours of people living with dementia.*
(Lismore/Northern Rivers)

Others noted the potential of innovative delivery methods, such as experiential learning or utilising technologies such as virtual reality or simulated learning. Collaboration was also seen as helping overcome logistical barriers by achieving the minimum number of participants for in-person training to be delivered.


*If people can’t travel because of being in a remote area, then have a ‘live’ training opportunity (virtual classroom) compared to something that is pre-recorded.*
(Lismore/Northern Rivers)

Appreciation for practical and individualised training immediately applicable to their field of practice was noted as a theme expressed by several participants.

#### 3.2.3. Personal Interest and Relevance

Across all roundtables, personal interests, including professional motivators, were identified as playing an important role in driving participation in dementia training. Incentives included professional benefits such as networking opportunities and gaining Continuing Professional Development (CPD) points. A strong motivator was having professional responsibilities linked to training, and some argued that training should also encompass values-based learning, fostering self-reflection on how individuals perceive people with dementia and influencing their overall approach to patient care.

Some comments reflected a distinction in attitudes and perceptions surrounding dementia training as opposed to other types of training. Perceptions towards dementia care can vary depending on a person’s role in engaging with people living with dementia and with attitudes depending on the work context (hospital or residential care):


*In a hospital setting, there is a stigma around health professionals working with people living with dementia as it is not an ‘attractive’ health area. If dementia can be framed with the idea there is hope around dementia care if you’re skilled… It is different in residential care because you work with people with dementia every day and improve life for residents.*
(Lismore)

The value of ongoing regular training, as opposed to one-off training, was widely noted, as was the importance of having the training delivered by an experienced trainer with expertise in dementia care. Participants reiterated across all sites that they wanted “*practical application to practice, such as strategies and problem-solving skills*” delivered by a trainer who was “*realistic and informed by the constraints in which staff are working*”.


*Practical and individualised training that is useful, realistic and can be applied to practice immediately following training, otherwise it is easily forgotten. For example, training that relates to processes within your working day… [and] provides tools to integrate changes of practice and review.*
(New Norfolk)

#### 3.2.4. Logistics and Convenience

Participants mentioned the importance of the convenience of training in their local area and having opportunities for training delivery that met their needs in terms of commitment given other demands on their time. Participants acknowledged the convenience of online training but noting that there were limitations, stating that “Some staff are tech savvy” and therefore able to do more online learning.

Participants also noted the importance of flexibility and being able to schedule online or short face-to-face training when there may be double staff (nursing and care staff).

#### 3.2.5. Costs

Financial and time costs were often expressed as factors affecting attending training, “*cost not only for training, but to have staff there for training*” and also often considerable travel time. It was also discussed that training is often mid-week and with some rural areas being some distance from training, this may lead to significantly longer periods away from work and home than just the training time and incur additional financial costs.

#### 3.2.6. Computers and Technology

Participants discussed technical difficulties with cameras or microphones and limited access to IT equipment at home. They noted that while online training at work is convenient, it can easily be disrupted if other work priorities arise, and there may be limited access to shared computers. Some also described internet connections as having “*hit & miss Wifi connectivity*” and being unreliable.


*IT issues with sound cutting in and out, when you can’t hear facilitators, and they can’t hear you can be distracting.*
(Carnarvon)

Others described different challenges with computer access and proficiency.


*Not every facility has access to a good computer. Computer literacy of some staff can also be an issue and some staff are not comfortable doing online training.*
(Chinchilla)

Others noted the benefits of technology make access to training easier and that online training can be more accessible.

### 3.3. Workplace and Training (Meso)

Themes that reflected the workplace, organisation or team level included human resources, competing workplace training priorities, the values and learning culture of the workplace, costs, and opportunities for translating knowledge from training into practice. In addition, there were factors related to the training organisation and the delivery of training in terms of relevance and fitness of the training to the organisations and culturally safe training design and delivery.

#### 3.3.1. Workplace Influences

##### Human Resources

Work shortages and high turnover notoriously impact the health and aged care workforce in rural and remote Australia [15,16]. Participants’ comments concerning high staff turnover impeding access to dementia training reflected this; low staffing was identified repeatedly as a significant barrier to accessing training. Workforce shortages often meant staff were too time-poor to attend training, and organisations found it difficult and costly to find staff to cover shifts to enable others to attend training.

A recurring theme was that low staffing levels, either through limited resources or recruitment and retention issues, were dominant issues affecting training access. Participants described that employers have difficulty releasing staff as there can be limited options to provide cover.

Staff turnover was noted as an issue that impacts training. Where staff turnover is high, regular, scheduled training can be challenging to maintain. Providing relief staff to cover training absences, communicating well in advance about training opportunities, and encouraging attendance were seen as meaningful ways that organisations could support training opportunities.

##### Competing Workplace Training Priorities

The impact of competing priorities, including the mandatory requirement for training on other topics, emerged as an important consideration. For most HACWs, dementia training competes with other training and was often prioritised lower than training that is mandated. This was considered particularly problematic when staff initially started work at organisations, as mandatory training takes time.


*The focus on training by employers at the moment is what is necessary, and what will ‘keep the doors open’ as a result of the aged care reform.*
(Lismore/Northern Rivers)

The aged care and health sector was regarded as a crowded learning space, where people are asked to learn a lot (across all areas, not just dementia). Providers have their own preferences and needs regarding what they want. While it is acknowledged everyone needs up-to-date training, the dynamic of prioritising this is challenging for staff and employers.


*The frequency and inconsistency of encountering clients with dementia can impact an individual’s decisions to allocate professional development allowances to dementia-related training.*
(Port Lincoln)

##### Workplace Values and Learning Culture

The value of organisational support was raised throughout, with participants appreciating organisations that encouraged and supported their attendance at relevant training. The importance of a workplace culture that values training was highlighted, for example, where training opportunities are communicated well in advance, staff are delegated to coordinate the event, and appropriate staff cover, funding or reimbursements are provided. Participants felt that training should be regarded as an investment, recognising that the short-term costs of training (additional expenses, staff absences, lower work activity) could be outweighed by the long-term benefits. Collaborative and inclusive approaches were favoured.


*In our smaller communities in Western Downs it makes more since that if we are facilitating training that we invite EVERYONE. This works two fold: firstly, bringing community together for a consistent approach and secondly a multidisciplinary approach for staff learning together.*
(Chinchilla)

##### Costs

It was described that organising training in a rural or remote area can be very costly and that the return on investment is not always possible, particularly with limited numbers of HACWs who can attend. Several participants noted the potential benefits both in learning and in cost efficiency to have organisations collaborate to provide local training.

##### Opportunities for Learning Translation

Participants distinguished between simply attending training and putting learnings into practice. They valued organisational support and follow-up to ensure learnings could be implemented. Participants required “space to think” about what learnings are practical and when to incorporate them in the workplace. They were interested in setting goals and being accountable for changes over time. Having the organisation assign accountability to staff for implementing learnings was proposed to ensure that learning outcomes are realised, for example, through brief staff presentations at staff meetings on relevant learnings.


*If there is no follow-up from an employer to ensure training is implemented on the ground, then it is unlikely to get implemented. Implementation should span across all levels of the organisation.*
(New Norfolk)

A suggestion for overcoming these issues was to have onsite training that aligned with workers’ schedules, such as harnessing multidisciplinary team meetings to engage different skill sets. One suggestion to follow up with support from the organisation to embed what was learned at the training in the workplace. This could be achieved through shadow/buddy shifts.

### 3.4. Training Organisation and Delivery Influences

#### The Relevance and “Fit” of Training for the Organisation

A recurring theme was that training design, organisation and delivery must be tailored to the rural and remote needs and contexts. Participants appreciated practical and individualised training immediately applicable to their field of practice. The value of ongoing regular training, as opposed to one-off training, was widely noted, as was the importance of having the training delivered by an experienced trainer with expertise in dementia care. Participants reiterated across all sites that they wanted “*practical application to practice, such as strategies and problem-solving skills*” delivered by a trainer who was “*realistic and informed by the constraints in which staff are working*”.

Practical and individualised training that is useful, realistic and can be applied to practice immediately following training; otherwise, it is easily forgotten. For example, training that relates to processes within your working day… (and) provides tools to integrate changes of practice and review. (New Norfolk)

Asked about preferences for training, several participants recalled trainings they had attended that were ‘emotive’ or ‘immersive’, noting these as memorable and more valuable. For example, one participant recalled:


*…training with an immersive experience and being in the shoes of somebody with dementia. It provoked different emotions, such as complete fear, helplessness which aided in understanding the feeling of patients living with dementia. It helps explain some of the responsive behaviours of people living with dementia.*
(Northern Rivers)

Others noted the potential of innovative delivery methods, such as experiential learning or utilising technologies such as virtual reality or simulated learning. Collaboration was also seen as helping overcome logistical barriers by achieving the minimum number of participants for in-person training to be delivered.

Participants reinforced the importance of constraining the length and commitment required to complete training to ensure it was manageable logistically and financially. It was commented that lengthy training “*can be prohibitive*”, particularly for small workplaces. Some participants preferred training that involved limited preparatory work before attending training and wanted key facts “*instead of getting burdensome articles*”. The timing and day of the week of training were identified as important as it affected staff members’ ability to attend.

Many participants found the dementia training they had attended had been too broad to address local needs. All roundtable groups identified that organisations could do more to address the training needs of their diverse workforce and appreciated training that was adapted or designed for the rural and remote context. Participants reported they were reluctant to attend dementia training they considered ‘*repetitive*’ or ‘*boring*’, much preferring hands-on training relevant to their day-to-day work.


*If a training presentation starts following the ‘dementia-is-bad’ formula, which is already well known, it is easy to switch off.*
(New Norfolk)

### 3.5. Culturally Safe Training Design and Delivery

Participants highlighted that the realities of working in rural and remote contexts can bring specific challenges to ensuring that training needs in culturally safe care are met. For example, delivering culturally safe aged care in rural and remote areas may be particularly challenging given the typically smaller workforce.

Three specific aspects were identified in multiple roundtables: training was not First Nations specific, not culturally appropriate for other culturally diverse groups, or it focussed on a specific group of workers so that attendees from different backgrounds felt neglected. For First Nations Australians living with dementia in rural and remote areas, it was reported that access to culturally supportive dementia care is hindered either by low numbers of First Nations HACWs in the local workforce and by the need for older First Nations people to relocate away from family and Country if care is not available locally. Many residents in aged care facilities have culturally and linguistically diverse (CALD) backgrounds, and this can create challenges that the workforce is not trained to address. Many workers are also from CALD backgrounds, which can further magnify intercultural differences. Both older HACWs and some older people in rural areas living with dementia may not have been exposed to a great deal of diversity; this was identified as a potential cause of interpersonal problems that are challenging to manage.

Participants’ experience of cultural training was that it was either lacking altogether, generic and not tailored to the region and its local population, or not relevant to dementia care. Some participants reported they had received mandatory broad cultural training, which did not explore the specifics of certain cultures prevalent in the area. Cultural training had been provided to health professionals in one site but did not cover aged care or dementia. Participants identified that management lacking understanding of the value or importance of cultural training was a barrier to accessing cultural training.

The suggestions made for improving care by providing training in culturally safe dementia care highlighted the importance of consultation with the target group for training, the need to include local scenarios as case studies and collaboration with local networks. Collaboration in local networks could facilitate regional input into the delivery and content of training with the inclusion of local scenarios for discussion. Participants believed such relevance could help them to retain information. Collaboration could also help overcome logistical barriers by achieving the minimum number of participants for in-person training to be delivered.

### 3.6. Funding and Regulation (Macro)

While the roundtable groups were generally focussed on their individual and work contexts, several broad (macro) scale themes were apparent. These related to funding, the lack of mandated dementia training and the need for reliable internet for online dementia training.

#### 3.6.1. Funding for Dementia Training

Participants commented that funding for dementia training in rural and remote areas was required. They noted that the lack of dedicated funding for dementia training within organisations was a barrier to accessing training.


*People are interested in training but have limited funding for PD [professional development], if it isn’t the interest or an area you regularly work in, you might not choose to use the PD funding for dementia.*
(Port Lincoln)

It was recognised that a number of organisations such as Dementia Training Australia are funded to provide dementia training, but not all participants were aware of what was available and that much was available at no cost. Participants commented that funding for dementia training specifically for rural and remote areas was required and that the lack of dedicated funding allocated for dementia training within organisations was a barrier to accessing training.

#### 3.6.2. Mandatory Training

To increase the uptake of dementia training, it was suggested that government needed to mandate that dementia training was essential for those working in aged care as employers would have to support their workers’ attendance to learn about dementia and caring for people with dementia. However, they commented that it was essential that the training content was customisable and relevant for care in rural and remote areas.

#### 3.6.3. Internet Quality

While personal access to computers and technology was noted as a personal theme, variable access to high-quality internet connection was raised at only one remote roundtable.

## 4. Discussion

Training of health and aged care providers has been identified as a key pathway to improving the care received by and quality of life of people living with dementia [11,17,18]. Yet, surprisingly little research has focussed on the perspectives of the rural and remote HACW accessing dementia training [12]. Findings in this study, derived from a series of roundtable discussions, identified interrelated themes spanning from the individual worker to the government level.

Overall, our findings concur with issues identified in the literature. Participants identified barriers and facilitators to accessing training similar to those described previously in health-related contexts, including limited resources (lack of funding, staff and time) to facilitate investment in training [10,19], and a strongly expressed need for collaboration and shared ownership with users in development and implementation [20,21,22]. Barriers specific to rural and remote areas highlighted by participants were consistent with those identified in other studies. They included the costs of travel and distance, which make attending face-to-face training more of a challenge [9], and IT and internet issues [9,23]. It was frequently raised that dementia training is not mandatory [5,24] even for staff who work in residential aged care where the prevalence of dementia is over 50% [1,5].

There was considerable commonality in the issues raised by the respondents in all the sites. Interestingly, issues related to technology and its use emerged in all sites. Workforce and the costs of training were major issues in all sites, but particularly those in more remote settings. The need for the training to be inclusive across disciplinary backgrounds and different care settings was reiterated in all settings but the importance of it being delivered with awareness of the local context most strongly evident in more remote settings. Individual sites differed in local CALD and Indigenous population groups, but all sites were conscious of the need to deliver culturally safe care and were keen for training to help support them in this.

The value of ongoing organisational support for training was reiterated throughout the discussions, and it was emphasised that this must be accompanied by implementation and follow-up embedded in organisational structures. These principles are consistent with the Canadian findings from a survey of rural home care staff [3] and the recommendations of Bayly and colleagues [10]. It was evident that lack of organisational support and commitment can impede staff from knowing about training opportunities, attending training, and implementing in practice what they learned from training. Furthermore, training is often prioritised without consulting staff [25], is not ongoing [3], or is not relevant to their role [25].

Comments in relation to the general delivery of training resounded with many of the key recommendations of experts in the dementia education and training field. These include collaborating with HACWs to develop education suitable to their needs, providing ongoing organisational support for training, embedding in organisational structures, and matching training to the educational needs arising from specific job roles [8,20]. However, in relation to the last of these, our participants wanted training that was inclusive of multiple disciplines and job roles. There was a desire for locally relevant training content, and for workplaces to support opportunities for inter-organisational learning that facilitates local networking for collaborative development of training content. Learning from peers is a positive means of incorporating a ‘whole-person’ approach to dementia care, and of improving knowledge and confidence through self-directed learning [17]. Sass and colleagues also recommended tailored programmes designed collaboratively with clinical service providers, as well as bringing together an interdisciplinary mix of learners to enhance knowledge exchange. An integrated team is a key enabler for good dementia care with comprehensive approach needed to overcome barriers and improve access to and the quality of training in rural and remote areas [10].

Shortcomings were identified related to the availability and take-up of training around providing culturally appropriate care for both Aboriginal and Torres Strait Islander people and people from CALD backgrounds; this aligns with issues identified in the Royal Commission [5]. Participants’ insights reflected wider views that the Australian aged care system does not ensure culturally safe care [18,26]. There are emerging possibilities to guide the provision of better care for older Indigenous Australians in community and other health settings [27].

Participants regularly raised the issue that dementia training is not mandatory for all HACWs and therefore is not prioritised over other mandatory training. Notably, personal care workers make up the largest proportion of the direct care workforce in residential aged care (70%), but currently the minimum qualification for a personal care worker in Australia does not require any education regarding dementia [28,29]. This aligns with the recommendation made by the Royal Commission for all workers in aged care services to undertake regular dementia training, and to mandate minimum education standards and ongoing training [5]. One issue raised was the limited nature of engagement with people living with dementia for some workers who interact with people living with dementia occasionally but not regularly, highlighting that different workers will benefit from different types of training. For example, mandatory training to cover minimum levels of knowledge and skills may be more valuable for PCAs early in employment, given their entry-level requirements for training at entry into aged care. However, a holistic and collaborative approach was favoured, given time constraints. Importantly, colleagues were identified as important sources of knowledge and participants suggested a multidisciplinary approach to encourage in-house education. Formal efforts at peer-assisted learning have been successful and warrant further exploration for rural and remote contexts [17]. In sum, this means that a potential approach is for training to be targeted (to staff need and appropriate to local context, including the culture of residents) and be integrative with expert delivery at one point, sustained by internal staff learning at another, and be ongoing (or generative).

Staff shortages and the smaller workforce in rural and remote areas present difficulties for releasing staff for training, and turnover also reduces the cost-effectiveness of face-to-face training [3,28,30]. Rural and remote health services typically have a proportionately smaller workforce than services in metropolitan areas, and previous studies have noted the difficulties in absorbing staff absences to attend training and the direct impact on the provision of care [9]. Participants at all roundtables commented on low staffing levels impeding attendance at dementia training; in rural Canada, this was identified as the top challenge in accessing dementia training regardless of professional role [31]. This situation creates a vicious cycle where shortages reduce training opportunities, but the reduced training opportunities (and subsequent care quality) increase staff turnover. So, prioritising training may be ‘sold’ to organisations as a crucial first step in staff retention and quality care delivery. It is essential to reiterate that these issues cannot be addressed without ongoing efforts to improve quality, personalised dementia care in rural aged care homes through emphasis on staff retention and recruitment and resource adequacy [3].

Participant comments, although generally preferencing face-to-face training, reflected a desire for the flexibility provided by online training and a wish to have this option additionally available. This desire has been reflected in studies reporting that the ability to use an online platform rather than physically travelling to training is highly valued by staff in rural and remote areas [9,23,30]. Although poor-quality internet connection has been reported as a barrier in previous studies [31] and although slow internet speeds and unstable connections are notoriously problematic in many remote communities [32], participants were more likely to raise technical difficulties and hardware problems (dysfunctional cameras or microphones) or limited access to IT equipment at home as issues rather than internet connections per se. Accessing equipment and timely technology (IT) support for malfunctioning IT equipment may be more challenging or costly in rural and remote areas but is a particularly important consideration for dementia training and care in rural contexts.

Some recent studies are positive concerning the effectiveness of online training and provide evidence for increasing uptake. Williams and colleagues reported that 60% of dementia-specific education completed by residential aged care staff in Victorian residential aged care in the past two years was delivered online, and a small proportion also included an element of in-person delivery [30]. This is markedly higher than the recent data for work-related learning in Australia, where approximately 19% of work-related learning across diverse industries was completed online [30,31]. As the aged care sector has an older workforce compared with other industries (median age 46 years), this dispels the stereotype that older workers are less tech-savvy, unlikely to engage in online learning and have more difficulty learning new things and complex tasks. While HACW may prefer face-to-face training, for example, GPs in regional, rural and remote areas were more likely than those in urban areas to attend face-to-face workshops [9], workers will take up online learning opportunities if they meet their needs. The need to increase awareness of available online dementia training, such as that offered by Dementia Training Australia, and to encourage its use was also evident in the roundtables. Through collaborations, this could be linked with local information on pathways related to local dementia management [32].

Our findings emphasise the importance of ensuring equitable access to dementia training with a design that meets the needs and context of the rural workforce. As a result of these roundtable discussions, a key list of recommendations for how rural employers, HACW and governing bodies can work to improve the access and quality of dementia training for HACW in rural and remote areas and the care for those living with dementia is shown in Box 1.

Box 1Key Recommendations to Improve Dementia Training Uptake and Effectiveness for HACW in Rural and Remote Areas.Determine a minimum standard of dementia training for rural and remote HACW;Offer flexibility in modes of delivery given different learning needs;Ensure culturally responsive training is available to all rural and remote HACW;Co-design of training with the intended users, including collaboration with the intended recipients;Reflect rural and remote contexts in training design and delivery;Plan training well in advance of face-to-face training delivery;Use rural and remote-based key workers and experts to deliver dementia training programs to HACW in rural and remote areas;Ensure equitable access to training and support for rural areas, using a range of modalities including face-to-face and online training;Ensure rural and remote dementia training has a multidisciplinary approach and is offered across sites to allow rural teams to attend together and ensure training provides better outcomes for the money spent.

More effort needs to go to codesign of training with rural and remote HACWs, recognising and developing the knowledge and experience that exists in rural and remote areas so that training is infused with relevant local cultural insights and practices and relevant cultural awareness is included in all aspects of training. Flexible models of training are needed to meet broad and diverse rural and remote workforce needs as well as the rural and remote context. The capacity, expertise and experience of rural and remote HACW to develop solutions and strategies that facilitate and encourage HACW to share their knowledge should be recognised; training should support a team-based approach. There is also the opportunity to ensure cost efficiencies in training by multiple organisations collaborating to ensure a larger attendance in training programs offered for HACW in their region. Broader industry-based strategies also require consideration, resourcing rural and remote areas in staffing and through training of HACW and support for high quality dementia care for older people.

### Strengths and Limitations

Strengths of this study include the diversity of participants and sites across rural and remote locations in Australia. Participants were drawn from a variety of organisational types (public, private, non-government organisations) that provided a range of services in different settings to people with dementia across rural and remote Australia. As a result, our data provide a range of perspectives relating to barriers and enablers concerning the uptake of dementia training in rural and remote locations. The facilitated discussion format of the roundtables, incorporating concurrent clarification and analysis and follow-up member checking with opportunity to comment on the site report, helped ensure robust findings for important issues related to training in their regions.

A limitation may have arisen from the online format of some of the roundtables, which could have constrained some of the freedom of dialogue possible with entirely face-to-face meetings. However, smaller breakout rooms encouraged deliberations. A limited number of personal care workers and First Nations Australians participated in most roundtables, perhaps reflecting some of the issues raised about time pressures and their relative lack of autonomy over their work. Also, participants were sent a brief synopsis of the scoping review beforehand which could have influenced the discussions, although participants generally confirmed the findings but quickly moved to reporting on their local experience.

## 5. Conclusions

Overall, the roundtables provided valuable insights into the challenges and opportunities related to accessing dementia training in rural and remote areas. The overarching themes identified indicate that a holistic approach is needed to overcome current barriers to dementia training facing HACW in these settings. Key enablers identified in this study include organisational willingness to cover the cost of training and training that everyone, no matter their role, can attend together. Participants across all roundtables emphasised the value of ‘hands-on’ and locally available training that is applicable to their scope of practice. At every roundtable, participants raised their preference for knowledgeable, experienced, and passionate trainers. Participants highly valued face-to-face training, while concomitantly noting the flexibility provided by online courses and the importance of organisational support and commitment in ensuring staff access and training implementation. Generally, the findings support the existing literature but go further in indicating consensus amongst HACW in these areas about the need to avoid low-value generic training and for the importance of tailoring training to the local region, particularly in relation to provision of culturally safe dementia care. These findings contribute towards identifying targeted solutions for developing better dementia training for rural and remote areas, and thereby improving health care for people living with dementia in these areas.

## Figures and Tables

**Table 1 geriatrics-10-00028-t001:** Health profession background of participants (all sites) working rurally and with people with dementia.

Health Profession Background	Number (%)
Nursing	27 (41%)
Allied Health Professionals	15 (23%)
Management/Administration	15 (23%)
Personal Care/Support Workers	6 (9%)
General Practitioner	1 (1%)
Not specified	2 (3%)
**Total**	**66 (100%)**

**Table 2 geriatrics-10-00028-t002:** Characteristics of participants (all sites) working rurally and with people with dementia.

	Number (%) with Experience Working	
Years	Rurally or Remotely	With People with Dementia
Not specified	7 (11%)	8 (12%)
<1	7 (11%)	6 (9%)
1 to 5	11 (17%)	10 (16%)
6 to 10	10 (15%)	14 (21%)
11 to 20	12 (18%)	14 (21%)
≥20	19 (28%)	14 (21%)
Total	66 (100%)	66 (100%)

**Table 3 geriatrics-10-00028-t003:** Summary of key themes from the Roundtable discussions.

Level of Themes	Themes
**Individual Considerations (Micro)**	Learning preferences and priorities Engagement and participation Personal interest and relevance Logistics and convenience costs Computers and technology
**Workplace and Training (Meso)**	Human resources Competing workplace training priorities Workplace Values and Learning Culture Costs Opportunities for translating knowledge from training into practice Training Relevance Culturally safe training design and delivery
**Funding and Regulation (Macro)**	Funding for dementia training Mandatory training Internet quality and connection

## Data Availability

Data are unavailable due to privacy restrictions. Further information may be available by contacting the corresponding author.

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
