# Peer review of "Engaging Health and Aged Care Workers in Rural and Remote Australia Around Factors Impacting Their Access to and Participation in Dementia Training"

_geriatrics, 2025, doi:10.3390/geriatrics10010028_

Round 1

Reviewer 1 Report

Comments and Suggestions for Authors

I would strongly encourage you to make recommended revisions. It is an interesting and timely study.

Author Response

see file attached

Reviewer 2 Report

Comments and Suggestions for Authors

The topic of dementia education for rural primary care and direct care staff is important.

The authors survey healthcare workers utilizing facilitated focus group roundtables.

The manuscript would be greatly improved by providing short tables: healthcare workers and disciplines, key issues, topics, and general outline of the roundtables and processes used.

The authors should describe how they collated the data from the focus groups and organized it into themes. A table of the major themes would also be helpful.

The narrative throughout the manuscript is very conversational. A more concise description of the results highlighted tables will help greatly to clarify both the results as well as the discussion.

As an implementation presentation, a table of lessons learned or implementation barriers would also be helpful.

The conclusion should be short with a concise listing of specific findings and recommendations for future iterations of the program.

Author Response

see attached response

Round 2

Reviewer 2 Report

Comments and Suggestions for Authors

The authors are in substantial compliance with reviewer recommendations.